# Development of the Sphenoid Sinus in Japanese Children: A Retrospective Longitudinal Study Using Three-Dimensional Computed Tomography

**DOI:** 10.3390/jcm11216311

**Published:** 2022-10-26

**Authors:** Masaaki Higashino, Susumu Abe, Masaki Sawada, Hiroshi Yamada, Yusuke Ayani, Shin-Ichi Haginomori, Ryo Kawata, Toshihiro Matsuoka, Go Nakai, Keigo Osuga, Eiji Tanaka

**Affiliations:** 1Department of Otorhinolaryngology, Head and Neck Surgery, Osaka Medical and Pharmaceutical University, Takatsuki 569-8686, Osaka, Japan; 2Department of Comprehensive Dentistry, Tokushima University Graduate School of Biomedical Sciences, Tokushima 770-8504, Tokushima, Japan; 3Yamada Orthodontic Office, Izumiotsu 595-0025, Osaka, Japan; 4Department of Diagnostic Radiology, Faculty of Medicine, Osaka Medical and Pharmaceutical University, Takatsuki 569-8686, Osaka, Japan; 5Department of Orthodontics and Dentofacial Orthopedics, Tokushima University Graduate School of Biomedical Sciences, Tokushima 770-8504, Tokushima, Japan

**Keywords:** sphenoid sinus, cranial base, computed tomography, physical growth

## Abstract

Background: The sphenoid sinus (SS) is located close to vital structures, such as the pituitary gland, and it has significant clinical relevance. This study aimed to clarify the growth pattern of the SS in Japanese children using three-dimensional computed tomography (CT). Methods: Seventy-eight participants with congenital, acquired, or external auditory canal cholesteatoma were recruited and underwent CT more than twice during their treatment. Using the volume-rendered images, the size and volume of the SS were measured. Furthermore, on the scout image, the morphological measurements of the cranial base were determined. Results: The size and volume of the SS increased with age, and peaked at the mean age of 15 years. For males, the volume of the SS was smaller than that of females aged <5 years. The growth rate of the SS was significantly higher in males than in females. The maximum growth rate was detected at the age of 12 years for males and 10 years for females. For females, the increase in the length of the anterior cranial base ceased at approximately 10 years of age and remained constant thereafter. In contrast, for males, the length of the anterior cranial base increased gradually until 15 years of age. Conclusions: Considering the similarity of the periods between the adolescent growth spurt and the maximum growth rate of the SS, changes in the size of the SS may be used as an indicator of the physical growth spurt.

## 1. Introduction

The paranasal sinuses are filled with air and exist within the ethmoidal, frontal, maxillary, and sphenoidal bones [1]. All the sinuses are connected through a drain into the superior and lateral aspects of the nose [2], and the lining mucosa of the sinuses connects to the nasal cavity. The paranasal sinuses have complicated anatomy, with highly assorted shapes and sizes, and they command a considerable amount of space in the cranial bones. Studies that have examined the functions and factors affecting the size and shape of the sinuses reveal that the paranasal sinuses play an essential role in the immune defense and air filter system of the nose [3,4]. The cavity walls of the paranasal sinuses are gently coated with mucus, which keeps the tissue moist and healthy, and also acts as a bacterial trap.

The SS is the most concealed and least accessible of the four paranasal sinuses [5]. At birth, the SS is not fully developed (Shah et al., 2003). With pneumatization, the sinus expands in the anteroposterior and lateral directions and reaches the basisphenoid at 2 years [6,7]. By the age of 14 years, the sinus becomes fully developed, reaching its matured size [8,9], together with the other paranasal sinuses.

Since the SS is located adjacent to important structures, such as the optic nerve, internal carotid artery, second division of the trigeminal nerve, pituitary gland, and cavernous sinus [7,10], it is considered to be a structure of significant clinical relevance [11]. Furthermore, the SS is located in the center of the cranial base and neighbors the sella turcica, indicating its potential association with physical growth regulation. However, little information is available about the clinical impact of the SS.

Understanding the normal development of the SS is important for the accurate interpretation of diagnostic imaging regarding the abnormal development of SS. Thus, this study aimed to clarify the growth pattern of the SS in Japanese children and evaluate the impact of age and sex on the development of the SS using three-dimensional computed tomography (CT) performed longitudinally at the ages of 1–12 years.

## 2. Materials and Methods

### 2.1. Participants

Seventy-eight participants with congenital, acquired, or external auditory canal cholesteatoma, who were referred to the Otorhinolaryngology Department of the Osaka Medical and Pharmaceutical University Hospital from January 2010 to December 2021, were recruited for this study. The inclusion criteria for enrollment were patients with a diagnosis of congenital, acquired, or external auditory canal cholesteatoma using three-dimensional CT who were receiving treatment for congenital, acquired, or external auditory canal cholesteatoma in the hospital and undergoing more than two three-dimensional CT scans during their treatment. The exclusion criteria were a history of traumatic injury to the craniofacial region and systemic bone diseases, rhinosinusitis, hormonal disturbances, craniosynostosis, and neuropathic or neurological disorders.

This study was approved by the Ethics Committee of the Osaka Medical and Pharmaceutical University (approval no. 2021-143). Parental informed consent was obtained after the aims and procedures of the research were explained to the patients.

The present study estimated a sufficient size of the sample. The sample size was usually calculated using the effect size, power level, and significance level (Type I error). The effect size was adopted for convenient statistical parametric and non-parametric tests. To our knowledge, a suitable previous study could not be found for our effect size; therefore, the effect size in this study was thought of as medium (0.15). The statistical power (1-β) was computed with G*Power software (version 3.1.9.7; available from: https://www.psychologie.hhu.de/arbeitsgruppen/allgemeine-psychologie-und-arbeitspsychologie/gpower (accessed on 28 September 2022)). The power analysis consisted of a single regression analysis with an effect size of 0.15, a significance level of 0.05, and a power level of 0.8. Power analysis was conducted after the statistical analysis using a *post hoc* test. As a result, the total estimated sample size was decided to be 55 patients.

### 2.2. Three-Dimensional Computed Tomography

For the participants, three-dimensional CT was performed before the diagnosis of otitis media using a CT system (Canon Aquilion ONE TSX-301A/2A, Tokyo, Japan) with the following acquisition parameters: 120 kV auto exposure control mA; 0.5 mm collimation and 0.5 s rotation time; and 0.5 mm reconstruction thickness. CT was also performed after the treatment and during the follow-up period, with an interval between 1–2 years, to evaluate the treatment outcome of otitis media. Using a series of CT DICOM data from all the participants, a three-dimensional model and volume-rendered images of the SS were constructed (Figure 1). The SS volume was defined as an integrated value of the air cavity within the bony walls on the reformatted axial, sagittal, and coronal images. Volume-rendered images were adopted for the automatic calculation of the SS. The maximal width was defined as the distance between the most lateral points of the SS. The maximal height was defined as the distance from the top to the bottom of the SS. The maximal depth was defined as the distance between the most prominent points on the anterior and posterior parts of the SS. Moreover, based on a three-dimensional SS model, the morphological variations, including bilateral vs. unilateral and symmetric vs. asymmetric shapes, were evaluated. Based on the classification by previous studies [12,13,14], the extension was divided into the following types: conchal, presellar, sellar, and postsellar. Since the type of the extension can drastically change during growth, it was determined using CT images recorded at the end of the treatment.

Furthermore, using the scout image, the lengths of the anterior and posterior cranial base and the saddle angle (Nasion–Sella–Basion) were measured (Figure 2). Each scout image was traced onto an acetate paper by one investigator (M.S.). The tracings were computerized using a graphic digitizer (Dolphin Imaging, Dolphin Imaging & Management Solutions, Verona, Italy) by another investigator (M.H.) to measure the size and shape of the cranial base. The tracing precision was verified by two experts who joined this study as collaborators. All the examiners were blinded to the participants’ general condition. Before performing the analysis, the intra-examiner reliability of the measurements was determined using the intraclass correlation coefficient (ICC) on 20 randomly selected scout images that were traced and plotted using three arbitrary points (Nasion, Sella, and Basion points) by the same examiner twice within one week. As a result, the ICC was 0.990, verifying the reliability of the measurements.

### 2.3. Statistical Analysis

Statistical analyses were performed using SPSS version 27.0 (SPSS Inc., Chicago, IL, USA). The normality for age was assessed using the Shapiro–Wilk test. The age of the patient at the initial and last CT performance was calculated based on their demographic data. In addition, the difference in age between males and females was evaluated using unpaired comparisons between both groups (unpaired *t*-test or Mann–Whitney U test). A linear single regression analysis was performed to determine the relationships of the morphological variables (e.g., volume and diameter) of the SS to the age for each sex group. For single regression, the parallelism test was performed to compare the two groups. Furthermore, if the linear regression was parallel, the significance of the regression slope and the comparison of the average values between the two groups were determined. The α value was set at 0.05. *p* values less than 0.05 were considered statistically significant.

## 3. Results

### 3.1. Participants

The total sample size in this study was 78, including 45 males and 33 females, indicating that the sample size in this study was appropriate. All the participants in this study had undergone more than two three-dimensional CT scans for the diagnosis and treatment of congenital, acquired, or external auditory canal cholesteatoma.

The male patients had their initial CT scan taken at the mean age of 6.90 ± 2.75 years (mean and a standard deviation), and their ages ranged from 1 year to 12 years. The female patients had their initial CT scan at the mean age of 7.54 ± 3.45 years, and their ages ranged from 2 years to 12 years. No significant difference was found between males and females in terms of the ages at which they underwent the initial CT scan (*p* = 0.071).

The male patients had their last CT scan at the mean age of 13.54 ± 3.15 years, and their ages ranged from 6 years to 20 years. The female patients had their last CT at the mean age of 13.63 ± 4.40 years, and their ages ranged from 4 years to 23 years. No significant difference was found between the males and females in terms of the ages at which they underwent the last CT scan (*p* = 0.926).

### 3.2. Volumetric and Dimensional Measurements of the Sphenoid Sinus According to Age and Its Relationship with the Cranial Base

Of the total 78 participants, most exhibited bilateral SS. Half of the patients had a symmetric SS, and the remaining half had an asymmetric SS (Table 1). More than 60% of the patients developed a postsellar extension at the end of their treatment. No patient was found to have a conchal extension.

The SS largely increased in size with increasing age and reached its greatest volume at the age of 15 years. For males, the volume of the SS was smaller than that of females aged <5 years (Figure 3). The growth rate of the SS was significantly higher (*p* < 0.001) in males than in females. However, for males, the maximum growth rate occurred at the age of 12 years, while for females, it occurred at the age of 10 years. The average volume of the SS at the age of 15 years was 11.3 ± 3.48 cm^3^ for males, which was considerably larger than that of females (5.92 ± 3.89 cm^3^).

Regarding the dimensional measurements of the SS, the depth, height, and width of the SS increased significantly (*p* < 0.001) with age and reached its maximum at the age of approximately 15 years, irrespective of sex. The rate of dimensional increases was significantly (*p* < 0.001) greater in males than in females (Figure 4a–c). In particular, the rate at which the depth increased was greater than the rate at which the width and height increased, regardless of sex. For males, the average depth, height, and width of the SS at the age of >15 years were 33.79 ± 5.15 mm, 27.57 ± 2.18 mm, and 44.10 ± 9.58 mm, respectively, larger than those for females’ results (27.70 ± 12.55 mm, 21.38 ± 7.50 mm, and 31.35 ± 10.08 mm, respectively).

For the cranial base measurements, the cranial base angle (Ba-S-Na) was almost constant from 1 year to 23 years for both males and females (Figure 5a). The average values of the cranial base angle were 128.57 ± 4.12° and 128.09 ± 5.31° in males and 133.71 ± 5.00° and 134.68 ± 4.21° in females at the ages of 5 years and 15 years, respectively. The length of the anterior cranial base (SN) increased with age, and the rate of the increase was significantly (*p* < 0.001) higher in males than in females (Figure 5b). For females, the increase in the length of the anterior cranial base ceased at the age of approximately 10 years and remained constant thereafter. In males, the anterior cranial base lengthened gradually until 15 years of age. The length of the anterior cranial base was 65.89 ± 2.23 mm and 62.43 ± 2.48 mm for males and females at the age of 15 years, respectively. Regarding the posterior cranial base (SBa), the lengths at the age of 5 years were 38.33 ± 3.09 mm for males and 36.2 ± 2.41 mm for females (Figure 5c). The posterior cranial base grew until 15 years of age. Moreover, at the age of 15 years, the length of the posterior cranial base reached 47.36 ± 4.23 mm in males and 44.00 ± 2.82 mm in females.

## 4. Discussion

In the present study, the three-dimensional size and volume of the SS were longitudinally measured using CT images taken during treatment for congenital, acquired, or external auditory canal cholesteatoma in children. The average volume of the SS at the ages of 5 years and 15 years were 1.52 ± 1.60 cm^3^ and 11.83 ± 3.48 cm^3^ for males and 1.25 ± 1.37 cm^3^ and 5.92 ± 3.89 cm^3^ for females, respectively. The average volume of the SS at the age of 15 years was nearly consistent with the adult measurements, as reported in previous studies [12,15]. Most previous studies measured the size and volume of the SS in cross-sectional samples [12,15]. To our knowledge, this is the first study to longitudinally investigate the morphometric features of the SS during growth.

Arai et al. [16] investigated the relationship between the insufficient development of mastoid air cells and the abnormal structure of the paranasal sinuses in patients with chronic otitis media and acquired ear cholesteatoma using CT images and indicated that the developmental insufficiency in SS might be induced by long-term pediatric rhinosinusitis due to chronic middle ear inflammation in childhood. Odat et al. [17] also demonstrated that children with chronic rhinosinusitis were likely to have a less pneumatized SS. In the present study, we retrospectively collected the CT data of participants with congenital, acquired, or external auditory canal cholesteatoma; however, not all the participants exhibited rhinosinusitis. Furthermore, as mentioned above, our results showed no incidences of SS developmental insufficiency.

Our results demonstrated that the size and volume of the SS increased with age, and the age-related differences were statistically significant. Oliveira et al. [18] studied 47 CT scans obtained from 27 females and 20 males, aged between 18 years and 86 years and found no linear correlation between the SS volume and the participants’ age. Özer et al. [12] also analyzed the CT scans of 144 patients without any sinus pathology and concluded that no significant correlation was found between the size and volume of the SS and the participants’ age. A possible explanation for this inconsistency is that our CT images were longitudinally obtained from patients with congenital, acquired, or external auditory canal cholesteatoma. Since the SS is highly variable [12,19], the individual differences in the size, volume, and shape of the SS are considerably large. These findings indicated that the size and volume of the SS increased with age, which may suggest that the development of the SS may be related to physical growth.

In Japan, almost all young people aged between 9 years and 15 years experience the adolescent growth spurt, which is a time of rapid physical changes [20,21]. Due to sex differences, the adolescent growth spurt commences at the ages of 11 years and 13 years in females and males, respectively [21]. This implies that the adolescent growth spurt in males starts approximately two years later than that in females. The present study demonstrated the normal development of the SS in children and found that the maximum growth rate for the SS volume occurred at the age of approximately 12 years in males. For females, the growth spurt began 2 years earlier than for males. Considering the similarity between the timing of the adolescent growth spurt and the maximum growth rate of the SS, changes in the size of the SS could be used as an indicator of the physical growth spurt. Further studies with larger sample sizes that use a serial record of body heights in children are warranted to confirm the potential role of the SS in physical growth.

Regarding the sex difference of the SS, Karakas and Kavakli [22] analyzed the CT scans of 91 participants, aged between 5 years and 55 years, and reported a similar mean volume of the SS in males (6.83 ± 3.73 cm^3^) and females (6.00 ± 3.02 cm^3^). Conversely, other studies have shown a sex difference in the size and volume of the SS [12,13,14,18].

Of the four paranasal sinuses, the frontal and maxillary sinuses have been used for sex determination. Kiran et al. [23] analyzed the frontal sinus using 216 lateral cephalograms and determined the sex with 67.59% accuracy. In particular, the width, length, and ratio of a maximum width to a maximum height of the frontal sinus were used. Uthman et al. [24] also estimated the sex with 74% accuracy using the width, length, and height of the maxillary sinus. On the contrary, Özer et al. [12] did not detect a significant difference in the depth and width of the SS for the sex estimation. In this study, we demonstrated statistically significant differences in the size and volume of the SS between males and females. Furthermore, as described above, the maximum growth rate of the SS was significantly higher in males than in females and occurred later in males (12 years of age) than females (10 years of age). Although the morphological features of the SS were not markedly different between males and females, further studies are needed to examine whether the morphological and dimensional features of the SS can be used for sex determination.

## 5. Conclusions

The present study reported the average size and volume of the SS in growing children using three-dimensional CT scans taken longitudinally. Irrespective of sex, the size and volume of the SS increased with age and reached its greatest values at the age of 15 years. The growth rate of the SS was also significantly higher in males than in females. Additionally, the SS appeared at the ages of 12 years in males and 10 years in females. The morphological and dimensional features of the SS may be used in the estimation of sex and age.

## Figures and Tables

**Figure 1 jcm-11-06311-f001:**
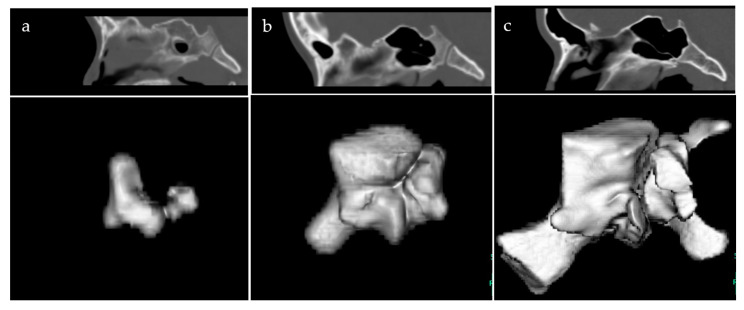
Representative images of the sphenoid sinus. (**a**) Presellar; (**b**) sellar; (**c**) postsellar.

**Figure 2 jcm-11-06311-f002:**
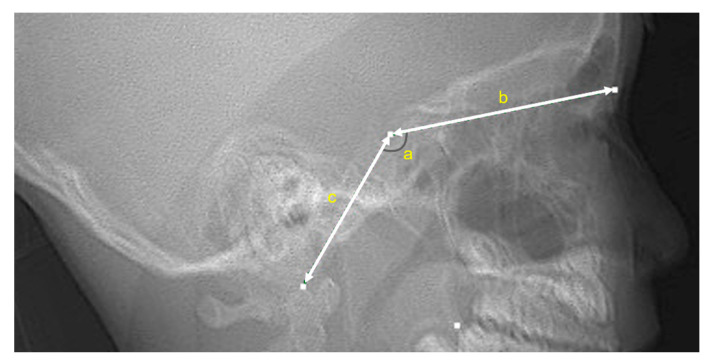
Representative scout image. (**a**) Cranial base angle; (**b**) anterior cranial base length; (**c**) posterior cranial base length.

**Figure 3 jcm-11-06311-f003:**
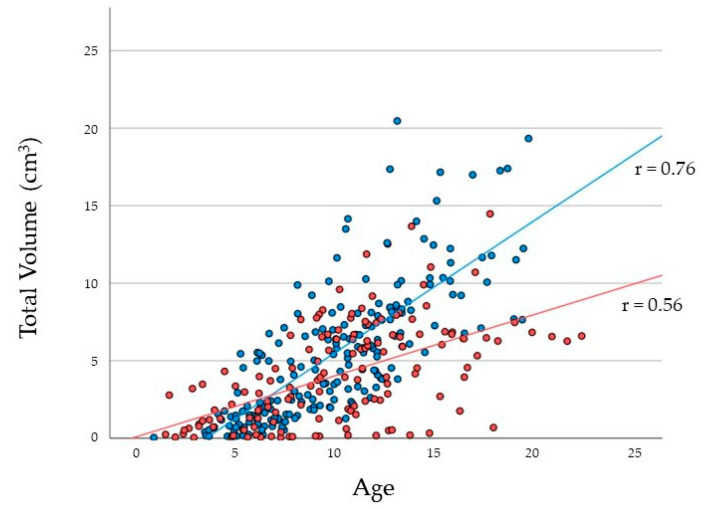
Scatter plots with linear single regression lines between the total volume of the sphenoid sinus at different ages for each sex. Red plots and lines indicate females, while blue plots and lines indicate males.

**Figure 4 jcm-11-06311-f004:**
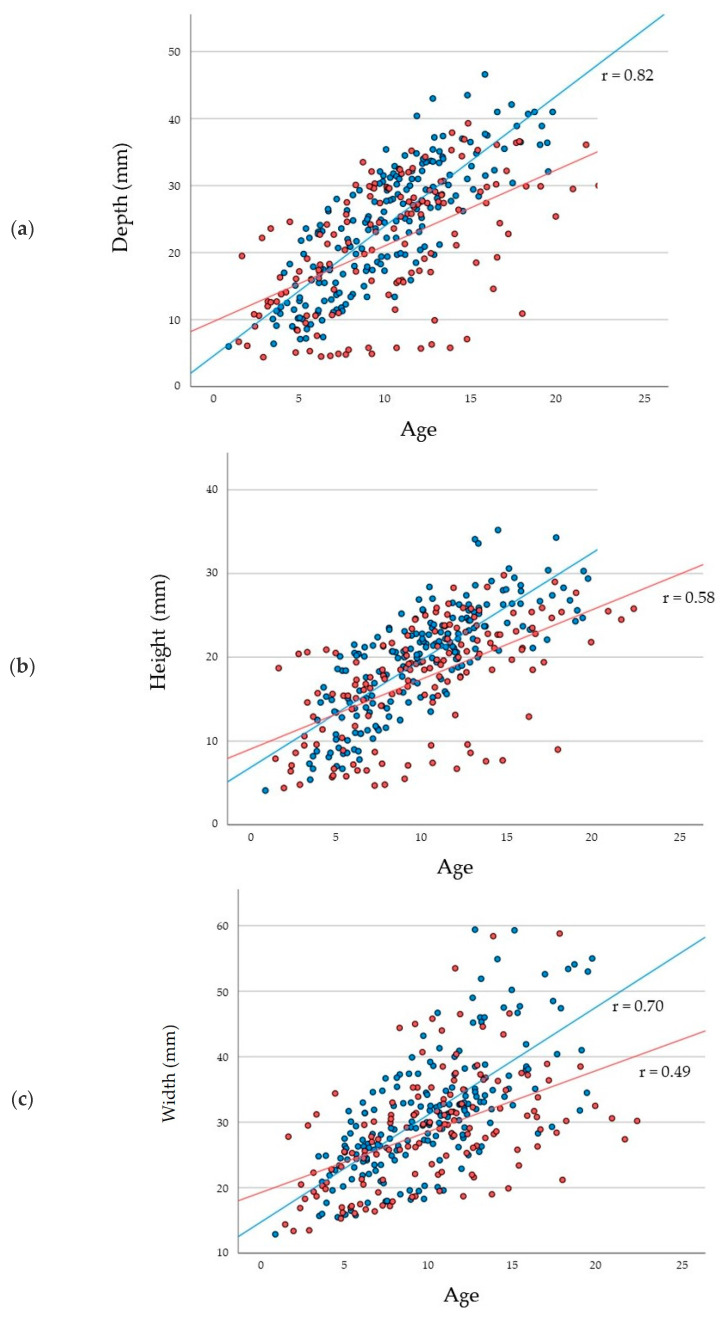
Scatter plots with linear single regression lines between the depth of the sphenoid sinus at different ages (**a**); between the height of the sphenoid sinus and age (**b**); and between the width of the sphenoid sinus and age (**c**) for each sex. Red plots and lines indicate females, while blue plots and lines indicate males.

**Figure 5 jcm-11-06311-f005:**
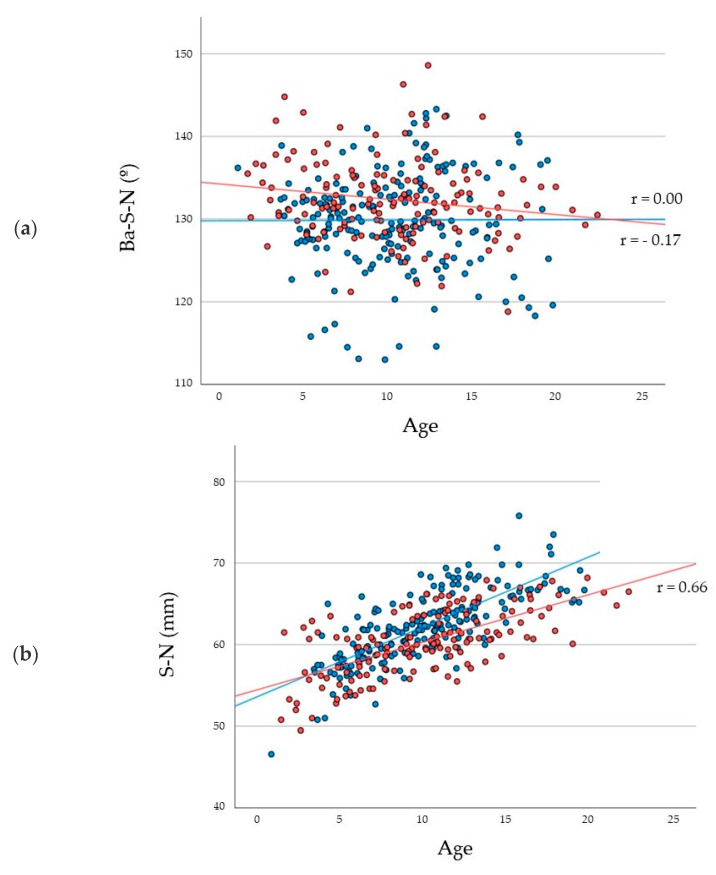
Scatter plots with linear single regression lines between the cranial base angle (Ba-S-N) and age (**a**); between the length of the anterior cranial base (S-N) and age (**b**); and between the length of the posterior cranial base and age (**c**) for each sex. Red plots and lines indicate females, while blue plots and lines indicate males.

**Table 1 jcm-11-06311-t001:** Morphological features of the sphenoid sinus.

	Males	Females	Total (%)
Bilateral or unilateral			
Bilateral	43	29	72 (92.3%)
Unilateral	2	4	6 (7.7%)
Symmetry or asymmetry			
Symmetry	22	14	36 (46.2%)
Asymmetry	23	19	42 (53.8%)
Type of extension			
Conchal	0	0	0 (0%)
Presellar	2	6	8 (10.3%)
Sellar	13	9	22 (28.2%)
Postsellar	30	18	48 (61.5%)

## Data Availability

Not applicable.

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
