# Peer review of "Development of the Sphenoid Sinus in Japanese Children: A Retrospective Longitudinal Study Using Three-Dimensional Computed Tomography"

_jcm, 2022, doi:10.3390/jcm11216311_

Round 1

Reviewer 1 Report

I congratulate for the well written manuscript! 

The study of "Development of Sphenoid Sinus in Japanese Children: A Lon-gitudinal Study Using 3-Dimensional Computed Tomography" aims to to determine the growth pattern of the sphenoid sinus in Japanese children and to assess the impact of age and sex on its development using computed tomography.

I may have some remarks:

-Revise Line 144 "...CTs tfor the diagnosis..."
-Revise the sentence of Line 154-155 "There was no significant difference in the ages between the males and females at which the last CT was (P = 0.926)."
-There is a contradiction between the text (Line 158) and Table1: in Table 1 6 Unilateral SS types was indicated, though in the text the following can be read: "Unilateral SS was not detected in any patient (Table 1)"

Author Response

Reviewer 1

I congratulate for the well written manuscript! 

(Response)

Thank you for your kind words.

The study of "Development of Sphenoid Sinus in Japanese Children: A Longitudinal Study Using 3-Dimensional Computed Tomography" aims to determine the growth pattern of the sphenoid sinus in Japanese children and to assess the impact of age and sex on its development using computed tomography.

I may have some remarks:

-Revise Line 144 "...CTs tfor the diagnosis..."

(Response)

Thank you for your suggestion. We have revised it. (revision: Line 164)

-Revise the sentence of Line 154-155 "There was no significant difference in the ages between the males and females at which the last CT was (P = 0.926)."

(Response)

Thank you for your suggestion. We have revised this sentence. (revision: Lines 173-175)

-There is a contradiction between the text (Line 158) and Table1: in Table 1 6 Unilateral SS types was indicated, though in the text the following can be read: "Unilateral SS was not detected in any patient (Table 1)"

(Response)

The reviewer is correct. We have deleted this sentence. (revision: Line 178)

Reviewer 2 Report

This study explores the development of sphenoid sinus in Japanese children with 3-deimenisonal CT and concluded that the change in size of the SS could be used as indicator of the physical growth spurt. The design is interesting and there are some clinical usages for clinical work. There are still some deficiencies in the manuscript. 

1.     Is this study a retrospective our prospective study? Please make it clear. 

2.     Why only chose cholesteatoma patients? Will this cause selective bias? 

3.     How to define the sufficient size of the study? 

4.     It is suggested to draw a schematic diagram to clearly show the saddle angle and cranial base. 

5.     “Unilateral SS was not detected in any patients” (Line 158). But there are 6 unilateral SS in Table1. 

6.     “The age of the patient at the initial CT and final CT in males and females was calculated.” How to calculate the age? What is the usage in the statistical analysis with the data? 

7.     There are some mistakes and should be checked and revised carefully. (Line 144, line 202) 

Author Response

Reviewer 2

This study explores the development of sphenoid sinus in Japanese children with 3-dimenisonal CT and concluded that the change in size of the SS could be used as indicator of the physical growth spurt. The design is interesting and there are some clinical usages for clinical work. There are still some deficiencies in the manuscript. 

  1. Is this study a retrospective our prospective study? Please make it clear. 

(Response)

Thank you for your suggestion. this is a retrospective study. To make it clear, we have added “a retrospective” in the title. (revision: Title)

  1. Why only chose cholesteatoma patients? Will this cause selective bias? 

(Response)

It is practically difficult to perform CT scans on healthy children on a regular basis due to the problem of radiation exposure. Therefore, in order to fully observe the changes of the sphenoid sinus with growth, we decided to retrospectively observe the changes of the sphenoid sinus using data from cases of cholesteatoma in which the temporal bone CT was regularly taken before and after the surgery at our department. (no revision)

  1. How to define the sufficient size of the study? 

(Response)

Thank you for your query. The sample size was usually calculated using the effect size, power and alpha value. To our knowledge, a suitable previous study could not find for the effect size, therefore, the effect size in this study was thought of as medium value (0.15). The sufficient sample size of this study was 55 patients. After statistical analysis, the adjusted R square was obtained for each single regression model. Since most these values were more than 0.4, we recalculated the required sample size using the adjusted R square as the effect size. The obtained sample size was 21 patients. As a result, the sample size of this study was considered sufficient for the statistical analyses. (revision: Lines 85-93)

  1. It is suggested to draw a schematic diagram to clearly show the saddle angle and cranial base. 

(Response)

Thank you for the excellent suggestion. We have added a schematic illustration for saddle angle and cranial base. (revision: Figure 2)

  1. “Unilateral SS was not detected in any patients” (Line 158). But there are 6 unilateral SS in Table1. 

(Response)

The reviewer is correct. We have deleted this sentence. (revision: Line 178)

  1. “The age of the patient at the initial CT and final CT in males and females was calculated.” How to calculate the age? What is the usage in the statistical analysis with the data? 

(Response)

Thank you for your query. the birthday and the date of CT performance of each patient were used as the demographic data. The ages were calculated as the years to be subtracted the date of CT performance from the birthday. The difference in age between male and female participants was evaluated using the Mann-Whitney U test, as these data were not normally distributed. (no revision)

  1. There are some mistakes and should be checked and revised carefully. (Line 144, line 202) 

(Response)

Thank you for your suggestion. We have checked and revised the manuscript thoroughly. (many revisions)

Reviewer 3 Report

This is a very interesting study which can facilitate understanding the development of sphenoid sinus and how chronic diseasescould influence this processComprehension of the anatomy of the sphenoid sinus is fundamental for a safesurgery.

would suggest to the Authors to betterexplain their choice to study children with cholesteatoma (and not healthy ones) and how this condition could impact on the development of the sphenoid sinus. In an article by Arai et al. (Arai Y, Sano D, Takahashi M, Nishimura G, Sakamaki K, Sakuma N, Komatsu M, Oridate N. Sphenoid sinus development in patientswith acquired middle ear cholesteatomaAuris Nasus Larynx. 2020 Jun;47(3):391-400) was noticed a developmentaldeficiency in sphenoid lenghtprobablycaused by middle ear inflammation in chronic rhinosinusitisAt the same time, asmentioned in the study by Odat et al. (OdatH, Almardeeni D, Tanash M, Al-Qudah M. Anatomical variation of the sphenoid sinusin paediatric patients and its associationwith age and chronic rhinosinusitis. J Laryngol Otol. 2019 Jun;133(6):482-486. doi: 10.1017/S0022215119000653. Epub2019 Apr 10. Erratum in: J Laryngol Otol. 2019 Aug;133(8):739), children with chronicrhinosinusitis have a less pneumatisedsphenoid sinusprobably because of an alteration of the normal conversion from redto yellow fatty bone marrow which normallyoccurs in the first two years of life.

Moreoverthis study does not cite the different types of sphenoid sinus (conchalpre sellar, sellar and post sellar): this isimportant for an adequate surgical planning. A study by Pirinc et al. (Pirinc B, Fazliogullari Z, Guler I, Unver Dogan N, Uysal II, Karabulut AK. Classification and volumetric study of the sphenoid sinus on MDCT images. Eur Arch Otorhinolaryngol. 2019 Oct;276(10):2887-2894) shows thatthe most common type of SS is sellar typein both sexesbut morphology and morphometry of the SS show individualdifferenceswhich are important for decision making and application for surgicalinterventions (especially transsphenoidalsurgery).

In the end, according to Our opinion, Authors should consider the differentanatomical types of SS and shoul considerthat chronic affections such ascholesteatoma could modify SS development in childrenso that volumes measured in this study might be smaller than in healthy controls

Author Response

Reviewer 3

This is a very interesting study which can facilitate under understanding the development of sphenoid sinus and how chronic diseases could influence this process. Comprehension of the anatomy of the sphenoid sinus is fundamental for a safe surgery.

I would suggest the authors to better explain their choice to study children with cholesteatoma (and not heal the ones) and how this condition could impact on the development of the sphenoid sinus. In an article by Arai Y, Sano D, Takahashi M, Nishimura G, Sakamaki K, Sakuma N, Komatsu M, Oridate N. Sphenoid sinus development in patients with acquired middle ear cholesteatoma. Auris Nasus Lanrynx 2020 Jun; 47(3):391-400) was noticed a developmental deficiency in sphenoid length, probably caused by middle ear inflammation in chronic rhinosinusitis. At the same time, as mentioned in the study by Odat et al. (Odat H, Almardeeni D, Tanash M, Al-Qudah M. Anatomical variation of the sphenoid sinus in paediatric patients and its association with age and chronic rhinosinusitis. J Laryngol Otol 2019; Jun;133(6):482-486. doi: 10.1017/S0022215119000653. Epub2019 Apr 10. Erratum in: J Laryngol Otol. 2019 Aug;133(8):739), children with chronic rhinosinusitis have a less pneumatized sphenoid sinus, probably because of an alteration of the normal conversion from red to yellow fatty bone marrow which normally occurs in the first two years of life.

Moreover, this study does not cite the different types of sphenoid sinus (conchal, presellar, sellar and postsellar): this is important for an adequate surgical planning. A study by Pirinc et al. (Princ B, Fazliogullari Z, Guter I, Unver Dogan N, Uysal II, Karabulut AK. Classification and volumetric study of the sphenoid sinus on MDCT images. Eur Arch Otorhinolaryngol. 2019 Oct;276(10):2887-2894) shows that the most common type of SS is sellar type in both sexes, but morphology and morphometry of the SS show individual differences, which are important for decision making and application for surgical interventions (especially transsphenoidal surgery).

In the end, according to our opinion, authors should consider the different anatomical types of SS and should consider that chronic affections such as cholesteatoma could modify SS development in children, so that volumes measured in this study might be smaller than in healthy controls.

 (Response)

Thank you for your suggestion. The reviewer is correct and the feedback is excellent and fruitful for us.

As the reviewer has pointed out, Arai et al. (2020) investigated the relationship between developmental insufficiency of mastoid air cells and abnormal morphology of the paranasal sinuses in patients with chronic otitis media (COM) and acquired middle ear cholesteatoma (AMEC) using CT images, and demonstrated that the developmental deficiency in sphenoid length caused by long-standing pediatric rhinosinusitis might indicate the existence of chronic middle ear inflammation in childhood and impact the pneumatization of mastoid air cells. We also studied children with congenital, acquired, or external auditory canal cholesteatoma as participants of this study, so that the data obtained in this study may not be regarded as control. However, our patients had no rhinosinusitis, and our result showed the average volume of the SS at the age of 15 years was nearly consistent with adult measurements as reported in previous studies. Therefore, we added the following paragraph in the Discussion section. In addition, several references were added according to the suggestion of the reviewer. (revisions: Lines 234-243; Ref # 14, 16, and 17)

Arai et al. [16] investigated the relationship between insufficient development of mastoid air cells and abnormal structure of the paranasal sinuses in patients with chronic otitis media and acquired ear cholesteatoma using CT images, and indicated that the developmental insufficiency in SS might be induced by long-term pediatric rhinosinusitis due to chronic middle ear inflammation in childhood. Moreover, Odat et al. [17] also demonstrated that children with chronic rhinosinusitis were likely to have a less pneumatized SS. In the present study, we retrospectively collected the CT data of participants with congenital, acquired, or external auditory canal cholesteatoma; however, all the participants did not exhibit rhinosinusitis. Furthermore, as mentioned above, our results showed no incidences of SS developmental insufficiency.

Reviewer 4 Report

As you mentioned, the growth of sphenoid sinus is related to the growth and development of body. Thus, in inclusion criteria, the mention on the overall physical growth and head growth should be included. Checking the head circumference using 3D CT will be needed.

Author Response

Reviewer 4

As you mentioned, the growth of sphenoid sinus is related to the growth and development of body. Thus, in inclusion criteria, the mention on the overall physical growth and head growth should be included. Checking the head circumference using 3D CT will be needed.

(Response)

Thank you for your suggestion. As we have described in the original manuscript, further studies with larger sample sizes using a serial record of body heights in children are warranted to confirm the potential role of the SS in physical growth (Lines 280-282). Unfortunately, this was a retrospective study, and we had no records of body heights of the participants. In addition, all the CTs were performed to diagnose and treat congenital, acquired, or external auditory canal cholesteatoma. Even if the patients suffered from otitis media, it is practically and ethically difficult to perform CT scans of the whole head on children on a regular basis due to the problem of radiation exposure. Therefore, we have regularly performed temporal bone CT by which we could observe not only the auris media but also the sphenoid sinus. According to the suggestion, we have added craniosynostosis as an exclusion criterion. (revision: Lines 80-81)

Round 2

Reviewer 2 Report

The author has answered the reviewer's queries satisfactorily and made appropriate modification.